# ONLINE LEARNING RATE ADAPTATION WITH HYPERGRADIENT DESCENT

**Atılım Güneş Baydin**
University of Oxford
gunes@robots.ox.ac.uk

**Robert Cornish**
University of Oxford
rcornish@robots.ox.ac.uk

**David Martínez Rubio**
University of Oxford
david.martinez2@wadham.ox.ac.uk

**Mark Schmidt**
University of British Columbia
schmidtm@cs.ubc.ca

**Frank Wood**
University of Oxford
fwood@robots.ox.ac.uk

## ABSTRACT

We introduce a general method for improving the convergence rate of gradient-based optimizers that is easy to implement and works well in practice. We demonstrate the effectiveness of the method in a range of optimization problems by applying it to stochastic gradient descent, stochastic gradient descent with Nesterov momentum, and Adam, showing that it significantly reduces the need for the manual tuning of the initial learning rate for these commonly used algorithms. Our method works by dynamically updating the learning rate during optimization using the gradient with respect to the learning rate of the update rule itself. Computing this "hypergradient" needs little additional computation, requires only one extra copy of the original gradient to be stored in memory, and relies upon nothing more than what is provided by reverse-mode automatic differentiation.

## 1 INTRODUCTION

In nearly all gradient descent algorithms the choice of learning rate remains central to efficiency; Bengio (2012) asserts that it is "often the single most important hyper-parameter" and that it always should be tuned. This is because choosing to follow your gradient signal by something other than the right amount, either too much or too little, can be very costly in terms of how fast the overall descent procedure achieves a particular level of objective value.

Understanding that adapting the learning rate is a good thing to do, particularly on a per parameter basis dynamically, led to the development of a family of widely-used optimizers including AdaGrad (Duchi et al., 2011), RMSProp (Tieleman & Hinton, 2012), and Adam (Kingma & Ba, 2015). However, a persisting commonality of these methods is that they are parameterized by a "pesky" fixed global learning rate hyperparameter which still needs tuning. There have been methods proposed that do away with needing to tune such hyperparameters altogether (Schaul et al., 2013) but their adoption has not been widespread, owing perhaps to their complexity, applicability in practice, or performance relative to the aforementioned family of algorithms.

Our initial conceptualization of the learning rate adaptation problem was one of automatic differentiation (Baydin et al., 2018). We hypothesized that the derivative of a parameter update procedure with respect to its global learning rate ought to be useful for improving optimizer performance. This conceptualization is not unique, having been explored, for instance, by Maclaurin et al. (2015). While the automatic differentiation perspective was integral to our conceptualization, the resulting algorithm turns out to simplify elegantly and not require additional automatic differentiation machinery. In fact, it is easily adaptable to nearly any gradient update procedure while only requiring one extra copy of a gradient to be held in memory and very little computational overhead; just a dot product in the

dimension of the parameter. Considering the general applicability of this method and adopting the name "hypergradient" introduced by Maclaurin et al. (2015) to mean a derivative taken with respect to a hyperparameter, we call our method *hypergradient descent*.

To our knowledge, our rediscovery appeared first in the largely neglected paper of Almeida et al. (1998), who arrived at the same hypergradient procedure as us. However, none of the aforementioned modern gradient-based optimization procedures existed at the time of its publication so the only examples considered were gradient and stochastic gradient descent on relatively simple functions. Having rediscovered this approach, we develop it further and demonstrate that adapting existing gradient descent procedures to use hypergradient descent to dynamically tune global learning rates improves stochastic gradient descent (SGD), stochastic gradient descent with Nesterov momentum (SGDN), and Adam; particularly so on large-scale neural network training problems.

For a given untuned initial learning rate, hypergradient algorithms consistently bring the loss trajectory closer to the optimal one that would be attained with a tuned initial learning rate, and thus significantly reduce the need for the expensive and time consuming practice of hyperparameter search (Goodfellow et al., 2016) for learning rates, which is conventionally performed using grid search, random search (Bergstra & Bengio, 2012), Bayesian optimization (Snoek et al., 2012), and model-based approaches (Bergstra et al., 2013; Hutter et al., 2013).

## 2 HYPERGRADIENT DESCENT

We define the hypergradient descent (HD) method by applying gradient descent on the learning rate of an underlying gradient descent algorithm, independently discovering a technique that has been previously considered in the optimization literature, most notably by Almeida et al. (1998). This differs from the reversible learning approach of Maclaurin et al. (2015) in that we apply gradient-based updates to a hyperparameter (in particular, the learning rate) at each iteration in an online fashion, instead of propagating derivatives through an entire inner optimization that consists of many iterations.

The method is based solely on the partial derivative of an objective function—following an update step—with respect to the learning rate. In this paper we consider and report the case where the learning rate $\alpha$ is a scalar. It is straightforward to generalize the introduced method to the case where $\alpha$ is a vector of per-parameter learning rates.

The most basic form of HD can be derived from regular gradient descent as follows. Regular gradient descent, given an objective function $f$ and previous parameters $\theta_{t-1}$, evaluates the gradient $\nabla f(\theta_{t-1})$ and moves against it to arrive at updated parameters

$$\theta_t = \theta_{t-1} - \alpha \nabla f(\theta_{t-1}) \,, \tag{1}$$

where $\alpha$ is the learning rate. In addition to this update rule, we would like to derive an update rule for the learning rate $\alpha$ itself. We make the assumption that the optimal value of $\alpha$ does not change much between two consecutive iterations so that we can use the update rule for the previous step to optimize $\alpha$ in the current one. For this, we will compute $\partial f(\theta_{t-1})/\partial \alpha$ , the partial derivative of the objective $f$ at the previous time step with respect to the learning rate $\alpha$. Noting that $\theta_{t-1} = \theta_{t-2} - \alpha \nabla f(\theta_{t-2})$, i.e., the result of the previous update step, and applying the chain rule, we get

$$\frac{\partial f(\theta_{t-1})}{\partial \alpha} = \nabla f(\theta_{t-1}) \cdot \frac{\partial(\theta_{t-2} - \alpha \nabla f(\theta_{t-2}))}{\partial \alpha} = \nabla f(\theta_{t-1}) \cdot (-\nabla f(\theta_{t-2})) \,, \tag{2}$$

which allows us to compute the needed hypergradient with a simple dot product and the memory cost of only one extra copy of the original gradient. Using this hypergradient, we construct a higher level update rule for the learning rate as

$$\alpha_t = \alpha_{t-1} - \beta \frac{\partial f(\theta_{t-1})}{\partial \alpha} = \alpha_{t-1} + \beta \nabla f(\theta_{t-1}) \cdot \nabla f(\theta_{t-2}) \,, \tag{3}$$

introducing $\beta$ as the hypergradient learning rate. We then modify Eq. 1 to use the sequence $\alpha_t$ to become

$$\theta_t = \theta_{t-1} - \alpha_t \nabla f(\theta_{t-1}) \,. \tag{4}$$

Equations 3 and 4 thus define the most basic form of the HD algorithm, updating both $\theta_t$ and $\alpha_t$ at each iteration. This derivation, as we will see shortly, is applicable to any gradient-based primal

**Algorithm 1** Stochastic gradient descent (SGD)

**Require:** $\alpha$: learning rate
**Require:** $f(\theta)$: objective function
**Require:** $\theta_0$: initial parameter vector
$\quad t \leftarrow 0$ ▷ Initialization
$\quad$ **while** $\theta_t$ not converged **do**
$\quad\quad t \leftarrow t + 1$
$\quad\quad g_t \leftarrow \nabla f_t(\theta_{t-1})$ ▷ Gradient
$\quad\quad u_t \leftarrow -\alpha\, g_t$ ▷ Parameter update
$\quad\quad \theta_t \leftarrow \theta_{t-1} + u_t$ ▷ Apply parameter update
$\quad$ **end while**
$\quad$ **return** $\theta_t$

**Algorithm 4** SGD with hyp. desc. (SGD-HD)

**Require:** $\alpha_0$: initial learning rate
**Require:** $f(\theta)$: objective function
**Require:** $\theta_0$: initial parameter vector
**Require:** $\beta$: hypergradient learning rate
$\quad t, \nabla_\alpha u_0 \leftarrow 0, 0$ ▷ Initialization
$\quad$ **while** $\theta_t$ not converged **do**
$\quad\quad t \leftarrow t + 1$
$\quad\quad g_t \leftarrow \nabla f_t(\theta_{t-1})$ ▷ Gradient
$\quad\quad h_t \leftarrow g_t \cdot \nabla_\alpha u_{t-1}$ ▷ Hypergradient
$\quad\quad \alpha_t \leftarrow \alpha_{t-1} - \beta\, h_t$ ▷ Learning rate update
$\quad\quad$ **Or, alternative to the line above:**
$\quad\quad \alpha_t \leftarrow \alpha_{t-1}\left(1 - \beta\, \dfrac{h_t}{\|g_t\|\,\|\nabla_\alpha u_{t-1}\|}\right)$ ▷ Mult. update
$\quad\quad u_t \leftarrow -\alpha_t\, g_t$ ▷ Parameter update
$\quad\quad \nabla_\alpha u_t \leftarrow -g_t$
$\quad\quad \theta_t \leftarrow \theta_{t-1} + u_t$ ▷ Apply parameter update
$\quad$ **end while**
$\quad$ **return** $\theta_t$

**Algorithm 2** SGD with Nesterov (SGDN)

**Require:** $\mu$: momentum
$\quad t, v_0 \leftarrow 0, 0$ ▷ Initialization
$\quad$ **Update rule:**
$\quad v_t \leftarrow \mu\, v_{t-1} + g_t$ ▷ "Velocity"
$\quad u_t \leftarrow -\alpha\,(g_t + \mu\, v_t)$ ▷ Parameter update

**Algorithm 5** SGDN with hyp. desc. (SGDN-HD)

**Require:** $\mu$: momentum
$\quad t, v_0, \nabla_\alpha u_0 \leftarrow 0, 0, 0$ ▷ Initialization
$\quad$ **Update rule:**
$\quad v_t \leftarrow \mu\, v_{t-1} + g_t$ ▷ "Velocity"
$\quad u_t \leftarrow -\alpha_t\,(g_t + \mu\, v_t)$ ▷ Parameter update
$\quad \nabla_\alpha u_t \leftarrow -g_t - \mu\, v_t$

**Algorithm 3** Adam

**Require:** $\beta_1, \beta_2 \in [0, 1)$: decay rates for Adam
$\quad t, m_0, v_0 \leftarrow 0, 0, 0$ ▷ Initialization
$\quad$ **Update rule:**
$\quad m_t \leftarrow \beta_1\, m_{t-1} + (1 - \beta_1)\, g_t$ ▷ 1st mom. estimate
$\quad v_t \leftarrow \beta_2\, v_{t-1} + (1 - \beta_2)\, g_t^2$ ▷ 2nd mom. estimate
$\quad \widehat{m}_t \leftarrow m_t/(1 - \beta_1^t)$ ▷ Bias correction
$\quad \widehat{v}_t \leftarrow v_t/(1 - \beta_2^t)$ ▷ Bias correction
$\quad u_t \leftarrow -\alpha\, \widehat{m}_t/(\sqrt{\widehat{v}_t} + \epsilon)$ ▷ Parameter update

**Algorithm 6** Adam with hyp. desc. (Adam-HD)

**Require:** $\beta_1, \beta_2 \in [0, 1)$: decay rates for Adam
$\quad t, m_0, v_0, \nabla_\alpha u_0 \leftarrow 0, 0, 0, 0$ ▷ Initialization
$\quad$ **Update rule:**
$\quad m_t \leftarrow \beta_1\, m_{t-1} + (1 - \beta_1)\, g_t$ ▷ 1st mom. estimate
$\quad v_t \leftarrow \beta_2\, v_{t-1} + (1 - \beta_2)\, g_t^2$ ▷ 2nd mom. estimate
$\quad \widehat{m}_t \leftarrow m_t/(1 - \beta_1^t)$ ▷ Bias correction
$\quad \widehat{v}_t \leftarrow v_t/(1 - \beta_2^t)$ ▷ Bias correction
$\quad u_t \leftarrow -\alpha_t\, \widehat{m}_t/(\sqrt{\widehat{v}_t} + \epsilon)$ ▷ Parameter update
$\quad \nabla_\alpha u_t \leftarrow -\widehat{m}_t/(\sqrt{\widehat{v}_t} + \epsilon)$

Figure 1: Regular and hypergradient algorithms. *Left-hand side:* SGD with Nesterov (SGDN) (Algorithm 2) and Adam (Algorithm 3) are obtained by substituting the corresponding initialization (red) and update (blue) statements into regular SGD (Algorithm 1). *Right-hand side:* Hypergradient variants of SGD with Nesterov (SGDN-HD) (Algorithm 5) and Adam (Adam-HD) (Algorithm 6) are obtained by substituting the corresponding statements into hypergradient SGD (SGD-HD) (Algorithm 4).

optimization algorithm, and is computation- and memory-efficient in general as it does not require any more information than the last two consecutive gradients that have been already computed in the base algorithm.

### 2.1 DERIVATION OF THE HD RULE IN THE GENERAL CASE

Here we formalize the derivation of the HD rule for an arbitrary gradient descent method. Assume that we want to approximate a minimizer of a function $f : \mathbb{R}^n \to \mathbb{R}$ and we have a gradient descent method with update rule $\theta_t = u(\Theta_{t-1}, \alpha)$, where $\theta_t \in \mathbb{R}^n$ is the point computed by this method at step $t$, $\Theta_t = \{\theta_i\}_{i=0}^t$ and $\alpha$ is the learning rate. For instance, the regular gradient descent mentioned above corresponds to an update rule of $u(\Theta_t, \alpha) = \theta_t - \alpha \nabla f(\theta_t)$.

In each step, our goal is to update the value of $\alpha$ towards the optimum value $\alpha_t^*$ that minimizes the expected value of the objective in the next iteration, that is, we want to minimize $\mathbb{E}[f(\theta_t)] = \mathbb{E}[f(u(\Theta_{t-1}, \alpha_t))]$, where the expectation is taken with respect to the noise produced by the estimator of the gradient (if we compute the gradient exactly then the noise is just 0). We want to update the

previous learning rate $\alpha_{t-1}$ so the new computed value, $\alpha_t$, is closer to $\alpha_t^*$. As we did in the example above, we could perform a step gradient descent, where the gradient is

$$\frac{\partial \mathbb{E}[f \circ u(\Theta_t, \alpha_t)]}{\partial \alpha_t} = \mathbb{E}\left[\nabla_\theta f(\theta_t)^\top \nabla_\alpha u(\Theta_{t-1}, \alpha_t)\right] = \mathbb{E}\left[\tilde{\nabla}_\theta f(\theta_t)^\top \nabla_\alpha u(\Theta_{t-1}, \alpha_t)\right] \quad (5)$$

where $\tilde{\nabla}_\theta f(\theta_t)$ is the noisy estimator of $\nabla_\theta f(\theta_t)$. The last equality is true if we assume, as it is usual, that the noise at step $t$ is independent of the noise at previous iterations.

However we have not computed $\theta_t$ yet, we need to compute $\alpha_t$ first. If we assume that the optimum value of the learning rate at each step does not change much across iterations, we can avoid this problem by performing one step of the gradient descent to approximate $\alpha_{t-1}^*$ instead. The update rule for the learning in such a case is

$$\alpha_t = \alpha_{t-1} - \beta \tilde{\nabla}_\theta f(\theta_{t-1})^\top \nabla_\alpha u(\Theta_{t-2}, \alpha_{t-1}) . \quad (6)$$

We call the previous rule, the additive rule of HD. However, (see Martínez (2017), Section 3.1) it is usually better for this gradient descent to set

$$\beta = \beta' \frac{\alpha_{t-1}}{\left\|\tilde{\nabla} f(\theta_{t-1})\right\| \left\|\nabla_\alpha u(\Theta_{t-2}, \alpha_{t-1})\right\|} \quad (7)$$

so that the rule is

$$\alpha_t = \alpha_{t-1} \left(1 - \beta' \frac{\tilde{\nabla} f(\theta_{t-1})^\top \nabla_\alpha u(\Theta_{t-2}, \alpha_{t-1})}{\left\|\tilde{\nabla} f(\theta_{t-1})\right\| \left\|\nabla_\alpha u(\Theta_{t-2}, \alpha_{t-1})\right\|}\right) . \quad (8)$$

We call this rule the multiplicative rule of HD. One of the practical advantages of this multiplicative rule is that it is invariant up to rescaling and that the multiplicative adaptation is in general faster than the additive adaptation. In Figure 2 we can see in black one execution of the multiplicative rule in each case.

Applying these derivation steps to stochastic gradient descent (SGD) (Algorithm 1), we arrive at the hypergradient variant of SGD that we abbreviate as SGD-HD (Algorithm 4). As all gradient-based algorithms that we consider have a common core where one iterates through a loop of gradient evaluations and parameter updates, for the sake of brevity, we define the regular algorithms with reference to Algorithm 1, where one substitutes the initialization statement (red) and the update rule (blue) with their counterparts in the variant algorithms. Similarly we define the hypergradient variants with reference to Algorithm 4. In this way, from SGD with Nesterov momentum (SGDN) (Algorithm 2) and Adam (Algorithm 3), we formulate the hypergradient variants of SGDN-HD (Algorithm 5) and Adam-HD (Algorithm 6).

In Section 4, we empirically demonstrate the performance of these hypergradient algorithms for the problems of logistic regression and training of multilayer and convolutional neural networks for image classification, also investigating good settings for the hypergradient learning rate $\beta$ and the initial learning rate $\alpha_0$. Section 5 discusses extensions to this technique and examines the convergence of HD for convex objective functions.

## 3 RELATED WORK

### 3.1 LEARNING RATE ADAPTATION

Almeida et al. (1998) previously considered the adaptation of the learning rate using the derivative of the objective function with respect to the learning rate. Plagianakos et al. (2001; 1998) proposed methods using gradient-related information of up to two previous steps in adapting the learning rate. In any case, the approach can be interpreted as either applying gradient updates to the learning rate or simply as a heuristic of increasing the learning rate after a "successful" step and decreasing it otherwise.

Similarly, Shao & Yip (2000) propose a way of controlling the learning rate of a main algorithm by using an averaging algorithm based on the mean of a sequence of adapted learning rates, also investigating rates of convergence. The stochastic meta-descent (SMD) algorithm (Schraudolph et al.,

2006; Schraudolph, 1999), developed as an extension of the gain adaptation work by Sutton (1992), operates by multiplicatively adapting local learning rates using a meta-learning rate, employing second-order information from fast Hessian-vector products (Pearlmutter, 1994). Other work that merits mention include RPROP (Riedmiller & Braun, 1993), where local adaptation of weight updates are performed by using only the temporal behavior of the gradient's sign, and Delta-Bar-Delta (Jacobs, 1988), where the learning rate is varied based on a sign comparison between the current gradient and an exponential average of the previous gradients.

Recently popular optimization methods with adaptive learning rates include AdaGrad (Duchi et al., 2011), RMSProp (Tieleman & Hinton, 2012), vSGD (Schaul et al., 2013), and Adam (Kingma & Ba, 2015), where different heuristics are used to estimate aspects of the geometry of the traversed objective.

## 3.2 HYPERPARAMETER OPTIMIZATION USING DERIVATIVES

Previous authors, most notably Bengio (2000), have noted that the search for good hyperparameter values for gradient descent can be cast as an optimization problem itself, which can potentially be tackled via another level of gradient descent using backpropagation. More recent work includes Domke (2012), where an optimization procedure is truncated to a fixed number of iterations to compute the gradient of the loss with respect to hyperparameters, and Maclaurin et al. (2015), applying nested reverse automatic differentiation to larger scale problems in a similar setting.

A common point of these works has been their focus on computing the gradient of a validation loss at the end of a regular training session of many iterations with respect to hyperparameters supplied to the training in the beginning. This requires a large number of intermediate variables to be maintained in memory for being later used in the reverse pass of automatic differentiation. Maclaurin et al. (2015) introduce a reversible learning technique to efficiently store the information needed for exactly reversing the learning dynamics during the hyperparameter optimization step. As described in Sections 1 and 2, the main difference of our method from this is that we compute the hypergradients and apply hyperparameter updates in an online manner at each iteration,[1] overcoming the costly requirement of keeping intermediate values during training and differentiating through whole training sessions per hyperparameter update.

## 4 EXPERIMENTS

We evaluate the behavior of HD in several tasks, comparing the behavior of the variant algorithms SGD-HD (Algorithm 4), SGDN-HD (Algorithm 5), and Adam-HD (Algorithm 6) to that of their ancestors SGD (Algorithm 1), SGDN (Algorithm 2), and Adam (Algorithm 3) showing, in all cases, a move of the loss trajectory closer to the optimum that would be attained by a tuned initial learning rate. The algorithms are implemented in Torch (Collobert et al., 2011) and PyTorch (Paszke et al., 2017) using an API compatible with the popular *torch.optim* package,[2] to which we are planning to contribute via a pull request on GitHub.

Experiments were run using PyTorch, on a machine with Intel Core i7-6850K CPU, 64 GB RAM, and NVIDIA Titan Xp GPU, where the longest training (200 epochs of the VGG Net on CIFAR-10) lasted approximately two hours for each run.

### 4.1 ONLINE TUNING OF THE LEARNING RATE

Figure 2 demonstrates the general behavior of HD algorithms for the training of logistic regression and a multi-layer neural network with two hidden layers of 1,000 units each, for the task of image classification with the MNIST database. The learning rate $\alpha$ is taken from the set of $\{10^{-1}, 10^{-2}, 10^{-3}, 10^{-4}, 10^{-5}, 10^{-6}\}$ and $\beta$ is taken as $10^{-4}$ in all instances.[3] We observe that for any given untuned initial learning rate, HD algorithms (solid curves) consistently bring the loss

---

[1] Note that we use the training objective, as opposed to the validation objective as in Maclaurin et al. (2015), for computing hypergradients. Modifications of HD computing gradients for both training and validation sets at each iteration and using the validation gradient only for updating $\alpha$ are possible, but not presented in this paper.

[2] Code will be shared here: `https://github.com/gbaydin/hypergradient-descent`

[3] Note that $\beta = 0.02$ is for the multiplicative example.

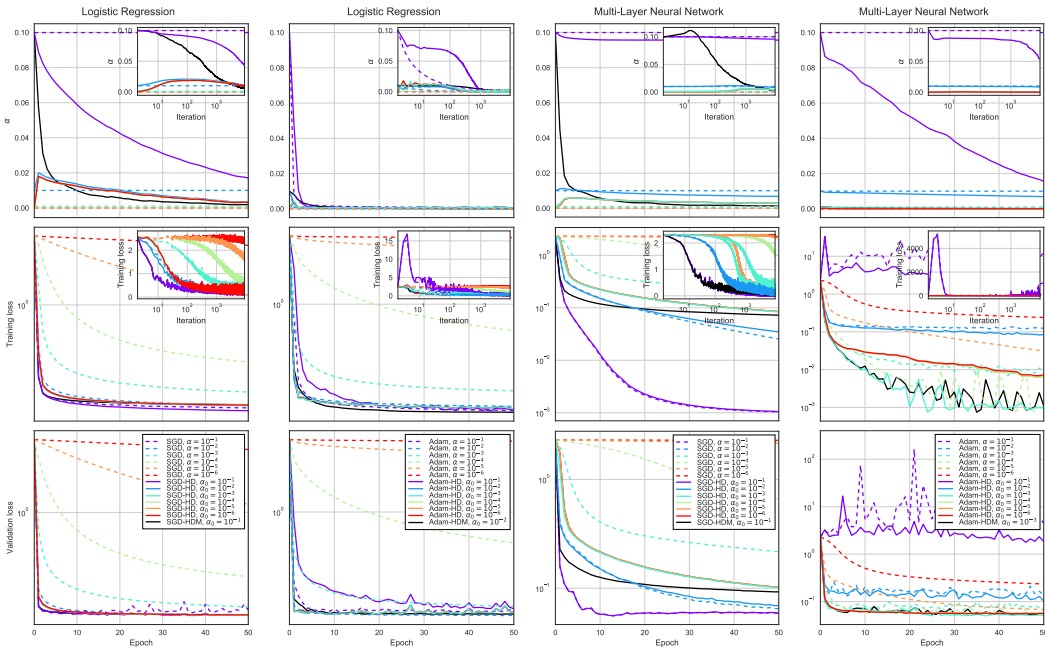

Figure 2: Online tuning of the learning rate for logistic regression and multi-layer neural network. Top row shows the learning rate, middle row shows the training loss, and the bottom row shows the validation loss. Dashed curves represent the regular gradient descent algorithms SGD and Adam, and solid curves represent their HD variants, SGD-HD and Adam-HD. HDM denotes an example of the multiplicative update rule.

trajectory closer to the optimal one that would be attained with the tuned initial learning rate of the non-HD algorithm (dashed curves).

In Figure 4 we report the results of a grid search for all the algorithms on the logitistic regression objective; similar results have been observed for the multi-layer neural network and CNN objectives as well. Figure 4 compels several empirical arguments. For one, independent of these results, and even if one acknowledges that using hypergradients for online learning rate adaption improves on the baseline algorithm, one might worry that using hypergradients makes the hyperparameter search problem worse. One might imagine that their use would require tuning both the initial learning rate $\alpha_0$ and the hypergradient learning rate $\beta$. In fact, what we have repeatedly observed and can be seen in this figure is that, given a good value of $\beta$, HD is somewhat insensitive to the value of $\alpha_0$. So, in practice tuning $\beta$ by itself, if hyperparameters are to be tuned at all, is actually sufficient.

Also note that in reasonable ranges for $\alpha_0$ and $\beta$, no matter which values of $\alpha_0$ and $\beta$ you choose, you improve upon the original method. The corollary to this is that if you have tuned to a particular value of $\alpha_0$ and use our method with an arbitrary small $\beta$ (no tuning) you will still improve upon the original method started at the same $\alpha_0$; remembering of course that $\beta = 0$ recovers the original method in all cases.

In the following subsections, we show examples of online tuning for an initial learning rate of $\alpha_0 = 0.001$, for tasks of increasing complexity, covering logistic regression, multi-layer neural networks, and convolutional neural networks.

### 4.1.1 Tuning Example: Logistic Regression

We fit a logistic regression classifier to the MNIST database, assigning membership probabilities for ten classes to input vectors of length 784. We use a learning rate of $\alpha = 0.001$ for all algorithms, where for the HD variants this is taken as the initial $\alpha_0$. We take $\mu = 0.9$ for SGDN and SGDN-HD. For Adam, we use $\beta_1 = 0.9$, $\beta_2 = 0.999$, $\epsilon = 10^{-8}$, and apply a $1/\sqrt{t}$ decay to the learning rate $\left(\alpha_t = \alpha/\sqrt{t}\right)$ as used in Kingma & Ba (2015) only for the logistic regression problem. We use the

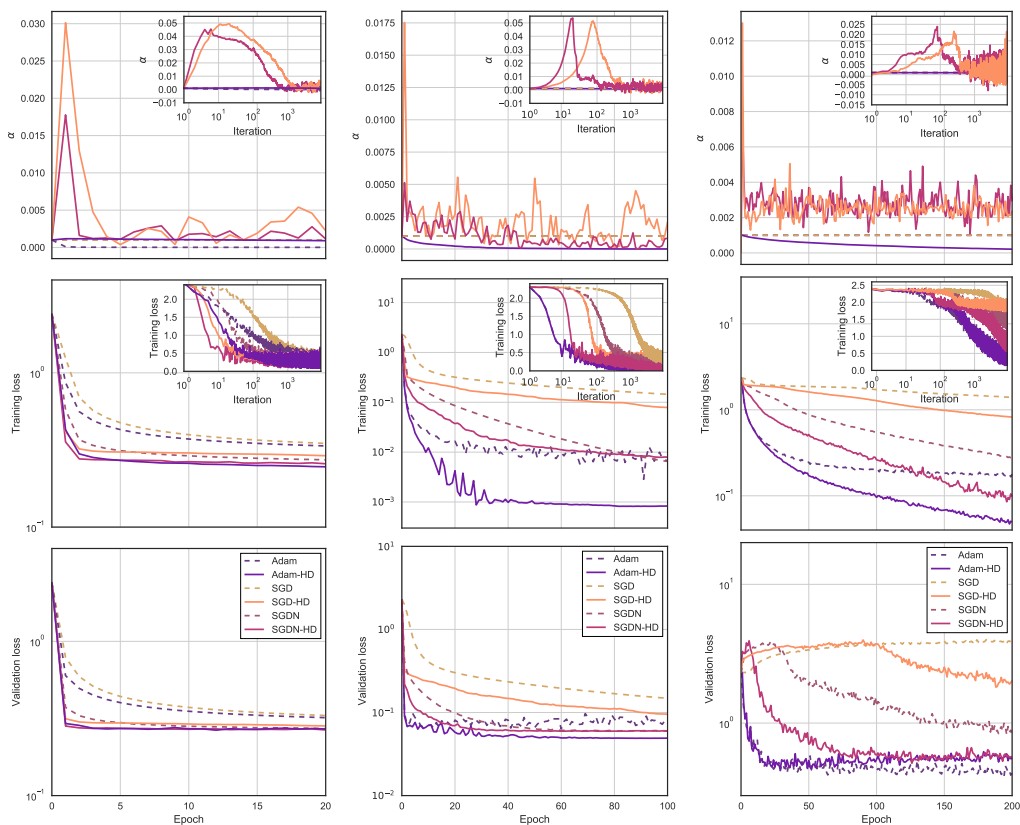

Figure 3: Behavior of hypergradient variants compared with their regular counterparts. Columns: *left:* logistic regression on MNIST; *middle:* multi-layer neural network on MNIST; *right:* VGG Net on CIFAR-10. Rows: *top:* evolution of the learning rate $\alpha_t$; *middle:* training loss; *bottom:* validation loss. Main plots show epoch averages and inset plots highlight the behavior of the algorithms during initial iterations. For MNIST one epoch is one full pass through the entire training set of 60,000 images (468.75 iterations with a minibatch size of 128) and for CIFAR-10 one epoch is one full pass through the entire training set of 50,000 images (390.625 iterations with a minibatch size of 128).

full 60,000 images in MNIST for training and compute the validation loss using the 10,000 test images. L2 regularization is used with a coefficient of $10^{-4}$. We use a minibatch size of 128 for all the experiments in the paper.

Figure 3 (left column) shows the negative log-likelihood loss for training and validation along with the evolution of the learning rate $\alpha_t$ during training, using $\beta = 0.001$ for SGD-HD and SGDN-HD, and $\beta = 10^{-7}$ for Adam-HD. Our main observation in this experiment, and the following experiments, is that the HD variants consistently outperform their regular versions.[4] While this might not come as a surprise for the case of vanilla SGD, which does not possess capability for adapting the learning rate or the update speed, the improvement is also observed for SGD with Nesterov momentum (SGDN) and Adam. The improvement upon Adam is particularly striking because this method itself is based on adaptive learning rates.

An important feature to note is the initial smooth increase of the learning rates from $\alpha_0 = 0.001$ to approximately 0.05 for SGD-HD and SGDN-HD. For Adam-HD, the increase is up to 0.001174 (a 17% change), virtually imperceivable in the plot due to scale. For all HD algorithms, this initial increase is followed by a decay to a range around zero. We conjecture that this initial increase and the later decay of $\alpha_t$, automatically adapting to the geometry of the problem, is behind the performance increase observed.

---

[4]We would like to remark that the results in plots showing loss versus training iterations remain virtually the same when they are plotted versus wall-clock time.

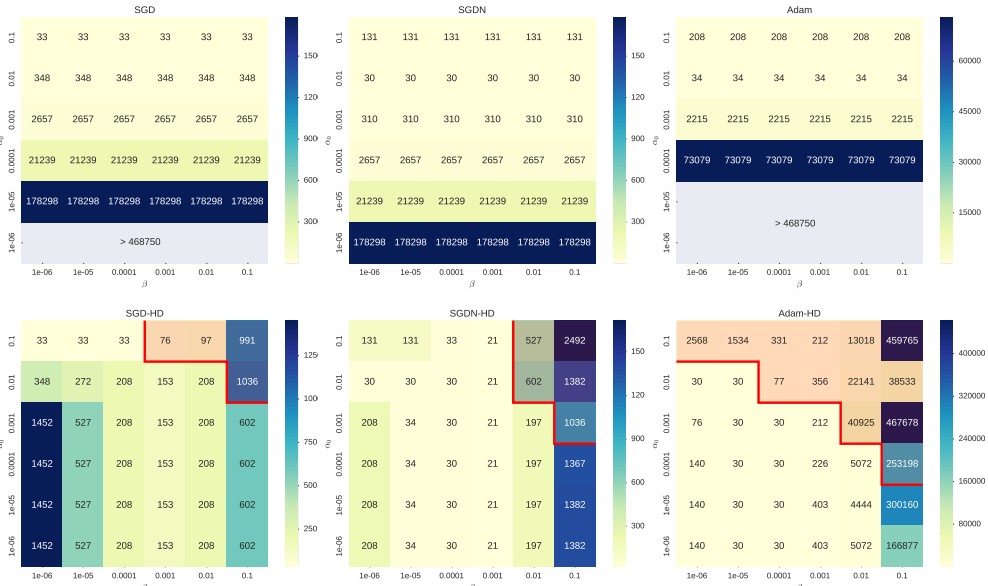

Figure 4: Grid search for selecting $\alpha_0$ and $\beta$, looking at iterations to convergence to a training loss of 0.29 for logistic regression. Everywhere to the left and below the shaded region marked by the red boundary, hypergradient variants (bottom) perform better than or equal to the baseline variants (top). In the limit of $\beta \to 0$, as one recovers the original update rule, the algorithms perform the same with the baseline variants in the worst case.

## 4.2 TUNING EXAMPLE: MULTI-LAYER NEURAL NETWORK

We next evaluate the effectiveness of HD algorithms on training a multi-layer neural network, again on the MNIST database. The network consists of two fully connected hidden layers with 1,000 units each and ReLU activations. We again use a learning rate of $\alpha = 0.001$ for all algorithms. We use $\beta = 0.001$ for SGD-HD and SGDN-HD, and $\beta = 10^{-7}$ for Adam-HD. L2 regularization is applied with a coefficient of $10^{-4}$.

As seen in the results in Figure 3 (middle column), the hypergradient variants again consistently outperform their regular counterparts. In particular, we see that Adam-HD converges to a level of validation loss not achieved by Adam, and shows an order of magnitude improvement over Adam in the training loss.

Of particular note is, again, the initial rise and fall in the learning rates, where we see the learning rate climb to 0.05 for SGD-HD and SGDN-HD, whereas for Adam-HD the overall behavior of the learning rate is that of decay following a minute initial increase to 0.001083 (invisible in the plot due to scale). Compared with logistic regression results, the initial rise of the learning rate for SGDN-HD happens noticeably before SGD-HD, possibly caused by the speedup from the momentum updates.

## 4.3 TUNING EXAMPLE: CONVOLUTIONAL NEURAL NETWORK

To investigate whether the performance we have seen in the previous sections scales to deep architectures and large-scale high-dimensional problems, we apply these to train a VGG Net (Simonyan & Zisserman, 2014) on the CIFAR-10 image recognition dataset (Krizhevsky, 2009). We base our implementation on the VGG Net architecture for Torch by Sergey Zagoruyko.[5] The network used has an architecture of (conv-64)×2 ∘ maxpool ∘ (conv-128)×2 ∘ maxpool ∘ (conv-256)×3 ∘ maxpool ∘ (conv-512)×3 ∘ maxpool ∘ (conv-512)×3 ∘ maxpool ∘ fc-512 ∘ fc-10, corresponding closely to the "D configuration" in Simonyan & Zisserman (2014). All convolutions have 3×3 filters and a padding of 1; all max pooling layers are 2×2 with a stride of 2. We use $\alpha = 0.001$ and $\beta = 0.001$ for SGD-HD and SGDN-HD, and $\beta = 10^{-8}$ for Adam-HD. We use the 50,000 training images in CIFAR-10 for training and the 10,000 test images for evaluating the validation loss.

---

[5] http://torch.ch/blog/2015/07/30/cifar.html

Looking at Figure 3 (right column), once again we see consistent improvements of the hypergradient variants over their regular counterparts. SGD-HD and SGDN-HD perform significantly better than their regular versions in the validation loss, whereas Adam and Adam-HD reach the same validation loss with relatively the same speed. Adam-HD performs significantly better than Adam in the training loss. For SGD-HD and SGDN-HD we see an initial rise of $\alpha$ to approximately 0.025, this rise happening, again, with SGDN-HD before SGD-HD. During this initial rise, the learning rate of Adam-HD rises only up to 0.001002.

## 5 CONVERGENCE AND EXTENSIONS

### 5.1 TRANSITIONING TO THE UNDERLYING ALGORITHM

We observed in our experiments that $\alpha$ follows a consistent trajectory. As shown in Figure 3, it initially grows large, then shrinks, and thereafter fluctuates around a small value that is comparable to the best fixed $\alpha$ we could find for the underlying algorithm without hypergradients. This suggests that hypergradient updates improve performance partially due to their effect on the algorithm's early behaviour, and motivates our first proposed extension, which involves smoothly transitioning to a fixed learning rate as the algorithm progresses.

More precisely, in this extension we update $\alpha_t$ exactly as previously via Eq. 8, and when we come to the update of $\theta_t$, we use as our learning rate a new value $\gamma_t$ instead of $\alpha_t$ directly, so that our update rule is $\theta_t = \theta_{t-1} + u(\Theta_{t-1}, \gamma_{t-1})$ instead of $\theta_t = \theta_{t-1} + u(\Theta_{t-1}, \alpha_{t-1})$ as previously. Our $\gamma_t$ satisfies $\gamma_t \approx \alpha_t$ when $t$ is small, and $\gamma_t \approx \alpha_\infty$ as $t \to \infty$, where $\alpha_\infty$ is some constant we choose. Specifically, $\gamma_t = \delta(t)\,\alpha_t + (1 - \delta(t))\,\alpha_\infty$ , where $\delta$ is some function such that $\delta(1) = 1$ and $\delta(t) \to 0$ as $t \to \infty$ (e.g., $\delta(t) = 1/t^2$).

Intuitively, this extension will behave roughly like HD at the beginning of the optimization process, and roughly like the original underlying algorithm by the end. We suggest choosing a value for $\alpha_\infty$ that would produce good performance when used as a fixed learning rate throughout.

Our preliminary experimental evaluation of this extension shows that it gives good convergence performance for a larger range of $\beta$ than without, and hence can improve the robustness of our approach. It also allows us to prove theoretical convergence under certain assumptions about $f$:

**Theorem 5.1.** *Suppose that $f$ is convex and $L$-Lipschitz smooth with $\|\nabla f(\theta)\| < M$ for some fixed $M$ and all $\theta$. Then $\theta_t \to \theta^*$ if $\alpha_\infty < 1/L$ and $t\,\delta(t) \to 0$ as $t \to \infty$, where the $\theta_t$ are generated according to (non-stochastic) gradient descent.*

*Proof.* Note that

$$|\alpha_t| \le |\alpha_0| + \beta \sum_{i=0}^{t-1} \left| \nabla f\left(\theta_{i+1}\right)^\top \nabla f\left(\theta_i\right) \right| \le |\alpha_0| + \beta \sum_{i=0}^{t-1} \|\nabla f\left(\theta_{i+1}\right)\| \|\nabla f\left(\theta_i\right)\| \le |\alpha_0| + t\beta M^2$$

where the right-hand side is $O(t)$ as $t \to \infty$. Our assumption about the limiting behaviour of $t\,\delta(t)$ then entails $\delta(t)\,\alpha_t \to 0$ and therefore $\gamma_t \to \alpha_\infty$ as $t \to \infty$. For large enough $t$, we thus have $1/(L+1) < \gamma_t < 1/L$, and the algorithm converges by the fact that standard gradient descent converges for such a (potentially non-constant) learning rate under our assumptions about $f$ (see, e.g., Karimi et al. (2016)). □

### 5.2 HIGHER-ORDER HYPERGRADIENTS

While our method adapts $\alpha_t$ during training, we still make use of a fixed $\beta$, and it is natural to wonder whether one can use hypergradients to adapt this value as well. To do so would involve the addition of an update rule analogous to Eq. 3, using a gradient of our objective function computed now with respect to $\beta$. We would require a fixed learning rate for this $\beta$ update, but then may consider doing hypergradient updates for *this* quantity also, and so on arbitrarily. Since our use of a single hypergradient appears to make a gradient descent algorithm less sensitive to hyperparameter selection, it is possible that the use of higher-order hypergradients in this way would improve robustness even further. We leave this hypothesis to explore in future work.

## 6 CONCLUSION

Having rediscovered a general method for adapting hyperparameters of gradient-based optimization procedures, we have applied it to the online tuning of the learning rate, and produced hypergradient descent variants of SGD, SGD with Nesterov momentum, and Adam that empirically appear to significantly reduce the time and resources needed to tune the initial learning rate. The method is general, memory and computation efficient, and easy to implement. The main advantage of the presented method is that, with a small $\beta$, it requires significantly less tuning to give performance better than—or in the worst case the same as—the baseline. We believe that the ease with which the method can be applied to existing optimizers give it the potential to become a standard tool and significantly impact the utilization of time and hardware resources in machine learning practice.

Our start towards the establishment of theoretical convergence guarantees in this paper is limited and as such there remains much to be done, both in terms of working towards a convergence result for the non-transitioning variant of hypergradient descent and a more general result for the mixed variant. Establishing convergence rates would be even more ideal but remains future work.

ACKNOWLEDGMENTS

Baydin and Wood are supported under DARPA PPAML through the U.S. AFRL under Cooperative Agreement FA8750-14-2-0006, Sub Award number 61160290-111668. Baydin is supported by the NVIDIA Corporation with the donation of the Titan Xp GPU used for this research. Cornish is supported by the EPSRC CDT in Autonomous Intelligent Machines and Systems. Martínez Rubio is supported by Intel BDC / LBNL Physics Graduate Studentship. Wood is supported by The Alan Turing Institute under the EPSRC grant EP/N510129/1; Intel; and DARPA D3M, under Cooperative Agreement FA8750-17-2-0093.

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
