# OpenReview forum: "Online Learning Rate Adaptation with Hypergradient Descent"
_ICLR.cc/2018/Conference — Accept (Poster)_

### Official Review · AnonReviewer3 · 2017-11-13
**good, but not perfect**

**Rating:** 6
**Confidence:** 4

**Review:**

SUMMARY:

The authors reinvent a 20 years old technique for adapting a global or component-wise learning rate for gradient descent. The technique can be derived as a gradient step for the learning rate hyperparameter, or it can be understood as a simple and efficient adaptation technique.


GENERAL IMPRESSION:

One central problem of the paper is missing novelty. The authors are well aware of this. They still manage to provide added value.
Despite its limited novelty, this is a very interesting and potentially impactful paper. I like in particular the detailed discussion of related work, which includes some frequently overlooked precursors of modern methods.


CRITICISM:

The experimental evaluation is rather solid, but not perfect. It considers three different problems: logistic regression (a convex problem), and dense as well as convolutional networks. That's a solid spectrum. However, it is not clear why the method is tested only on a single data set: MNIST. Since it is entirely general, I would rather expect a test on a dozen different data sets. That would also tell us more about a possible sensitivity w.r.t. the hyperparameters \alpha_0 and \beta.

The extensions in section 5 don't seem to be very useful. In particular, I cannot get rid of the impression that section 5.1 exists for the sole purpose of introducing a convergence theorem. Analyzing the actual adaptive algorithm would be very interesting. In contrast, the present result is trivial and of no interest at all, since it requires knowing a good parameter setting, which defeats a large part of the value of the method.


MINOR POINTS:

page 4, bottom: use \citep for Duchi et al. (2011).

None of the figures is legible on a grayscale printout of the paper. Please do not use color as the only cue to identify a curve.

In figure 2, top row, please display the learning rate on a log scale.

page 8, line 7 in section 4.3: "the the" (unintended repetition)

End of section 4: an increase from 0.001 to 0.001002 is hardly worth reporting - or am I missing something?

---

> ### Author Response · Authors · 2018-01-02
> **Thank you**
>
> > One central problem of the paper is missing novelty. The authors are well aware of this. They still manage to provide added value. Despite its limited novelty, this is a very interesting and potentially impactful paper.  I like in particular the detailed discussion of related work, which includes some frequently overlooked precursors of modern methods.
>
> Thank you very much for your evaluation and encouraging words.
>
> > The experimental evaluation is rather solid, but not perfect. It considers three different problems: logistic regression (a convex problem), and dense as well as convolutional networks. That's a solid spectrum. However, it is not clear why the method is tested only on a single data set: MNIST. Since it is entirely general, I would rather expect a test on a dozen different data sets. That would also tell us more about a possible sensitivity w.r.t. the hyperparameters \alpha_0 and \beta.
>
> Please note that we provide experimental evaluation on a non-MNIST data set, specifically CIFAR-10 (Section 4.3 on page 8 and Figure 2 on page 7).
>
> > The extensions in section 5 don't seem to be very useful. In particular, I cannot get rid of the impression that section 5.1 exists for the sole purpose of introducing a convergence theorem. Analyzing the actual adaptive algorithm would be very interesting. In contrast, the present result is trivial and of no interest at all, since it requires knowing a good parameter setting, which defeats a large part of the value of the method.
>
> We agree with your assessment that the analysis in Section 5.1 is significantly restricted and this is a limitation of the current paper. There remains much to be done in this respect, and a theoretical convergence analysis is a highly desired future work. Please note that a convergence analysis of the technique in the multidimensional quadratic case is available in a separate work, which we will highlight prominently in the de-anonymized final revision of the paper.
>
> > MINOR POINTS
>
> Thank you for pointing these out, we will fix them in the final revision.

---

### Official Review · AnonReviewer1 · 2017-11-17
**Somewhat weak novelty, but well written, complete, and potentially impactful.**

**Rating:** 7
**Confidence:** 3

**Review:**

The authors consider a method (which they trace back to 1998, but may have a longer history) of learning the learning rate of a first-order algorithm at the same time as the underlying model is being optimized, using a stochastic multiplicative update. The basic observation (for SGD) is that if \theta_{t+1} = \theta_t - \alpha \nabla f(\theta_t), then \partial/\partial\alpha f(\theta_{t+1}) = -<\nabla f(\theta_t), \nabla f(\theta_{t+1})>, i.e. that the negative inner product of two successive stochastic gradients is equal in expectation to the derivative of the tth update w.r.t. the learning rate \alpha.

I have seen this before for SGD (the authors do not claim that the basic idea is novel), but I believe that the application to other algorithms (the authors explicitly consider Nesterov momentum and ADAM) are novel, as is the use of the multiplicative and normalized update of equation 8 (particularly the normalization).

The experiments are well-presented, and appear to convincingly show a benefit. Figure 3, which explores the robustness of the algorithms to the choice of \alpha_0 and \beta, is particularly nicely-done, and addresses the most natural criticism of this approach (that it replaces one hyperparameter with two).

The authors highlight theoretical convergence guarantees as an important future work item, and the lack of them here (aside from Theorem 5.1, which just shows asymptotic convergence if the learning rates become sufficiently small) is a weakness, but not, I think, a critical one. This appears to be a promising approach, and bringing it back to the attention of the machine learning community is valuable.

---

> ### Author Response · Authors · 2018-01-02
> **Thank you**
>
> > I have seen this before for SGD (the authors do not claim that the basic idea is novel), but I believe that the application to other algorithms (the authors explicitly consider Nesterov momentum and ADAM) are novel, as is the use of the multiplicative and normalized update of equation 8 (particularly the normalization).
>
> > The experiments are well-presented, and appear to convincingly show a benefit. Figure 3, which explores the robustness of the algorithms to the choice of \alpha_0 and \beta, is particularly nicely-done, and addresses the most natural criticism of this approach (that it replaces one hyperparameter with two).
>
> Thank you very much for your evaluation and your encouraging feedback.
>
> Figure 3 was produced with exactly the purpose that you described, and we are very glad that this was noticed and found useful.
>
> > The authors highlight theoretical convergence guarantees as an important future work item, and the lack of them here (aside from Theorem 5.1, which just shows asymptotic convergence if the learning rates become sufficiently small) is a weakness, but not, I think, a critical one. This appears to be a promising approach, and bringing it back to the attention of the machine learning community is valuable.
>
> We agree that a theoretical convergence analysis is a highly desired future work and is a limitation of the current paper. We also agree with the assessment that the approach appears promising and therefore we would like to bring it to the attention of the larger community.

---

### Official Review · AnonReviewer2 · 2017-11-28
**interesting idea, but weak experiments**

**Rating:** 7
**Confidence:** 4

**Review:**


This paper revisits an interesting and important trick to automatically adapt the stepsize. They consider the stepsize as a parameter to be optimized and apply stochastic gradient update for the stepsize. Such simple trick alleviates the effort in tuning stepsize, and can be incorporated with popular stochastic first-order optimization algorithms, including SGD, SGD with Nestrov momentum, and Adam. Surprisingly, it works well in practice.

Although the theoretical analysis is weak that theorem 1 does not reveal the main reason for the benefits of such trick, considering their performance, I vote for acceptance. But before that, there are several issues need to be addressed.

1, the derivation of the update of \alpha relies on the expectation formulation. I would like to see the investigation of the effect of the size of minibatch to reveal the variance of the gradient in the algorithm combined with such trick.

2, The derivation of the multiplicative rule of HD relies on a reference I cannot find. Please include this part for self-containing.

3, As the authors claimed, the Maclaurin et.al. 2015 is the most related work, however, they are not compared in the experiments. Moreover, the empirical comparisons are only conducted on MNIST. To be more convincing, it will be good to include such competitor and comparing on practical applications on CIFAR10/100 and ImageNet.

Minors:

In the experiments results figures, after adding the new trick, the SGD algorithms become more stable, i.e., the variance diminishes. Could you please explain why such phenomenon happens?

---

> ### Author Response · Authors · 2018-01-02
> **Thank you**
>
> Thank you for your encouraging evaluation and for the improvements suggested.
>
> > 1, the derivation of the update of \alpha relies on the expectation formulation. I would like to see the investigation of the effect of the size of minibatch to reveal the variance of the gradient in the algorithm combined with such trick.
>
> We do not have theoretical results about the effect of the minibatch size and gradient variance on the hypergradient descent (HD) algorithm. Considering that the reviewer was potentially referring to experimental evidence, we will make sure to include experimental results with varying minibatch sizes in an appendix in the final revision of this paper.
>
> > 2, The derivation of the multiplicative rule of HD relies on a reference I cannot find. Please include this part for self-containing.
>
> Thank you for pointing this out. The mentioned reference for the multiplicative HD rule is now made accessible online, and can be located with a Google search of the title.
>
> > 3, As the authors claimed, the Maclaurin et.al. 2015 is the most related work, however, they are not compared in the experiments. Moreover, the empirical comparisons are only conducted on MNIST. To be more convincing, it will be good to include such competitor and comparing on practical applications on CIFAR10/100 and ImageNet.
>
> As you point out, Maclaurin et al. (2015) is a highly related work, which introduces the term “hypergradient” and similarly performs gradient-based updates of hyperparameters through a reversible higher-order automatic differentiation setup.
>
> However, note that in the approach in Maclaurin et al. (2015) a regular optimization procedure is truncated to a fixed number N of “elementary” iterations (such as N = 100 in the paper), at the end of which the derivative of an objective is propagated all the way through this N inner optimization iterations (the “reversibility” trick introduced in the paper is for making this possible in practice), and the resulting hypergradient is used in an outer optimization of M “meta” iterations (such as M=50 in the paper). Our technique, in contrast, is an online adaptation of a hyperparameter (in particular, the learning rate) at each iteration of optimization, and does not perform derivative propagation through an inner optimization that consists of many iterations. The techniques are thus not directly comparable as competing alternatives. For instance, it is not straightforward to replicate our learning rate trajectory through the VGGNet/CIFAR-10 experiment of 78125 iterations (Figure 2 on page 7, rightmost column) in the reversible learning algorithm due to (1) uninformative gradients beyond a few hundred iterations (see Section 4 “Limitations” in Maclaurin et al. 2015) and (2) potentially prohibitive memory requirements. Having said this, we believe that it would be interesting to compare the behavior of our algorithm for the initial 100 iterations with the 100-iteration learning-rate schedules reported in Maclaurin et al. (2015) and we intend to add such an experiment in the appendix in the final revision of the paper.
>
> > Moreover, the empirical comparisons are only conducted on MNIST.
>
> Please note that the paper does report non-MNIST empirical comparisons, specifically CIFAR-10 (Section 4.3 on page 8 and Figure 2 on page 7).
>
> > Minors: In the experiments results figures, after adding the new trick, the SGD algorithms become more stable, i.e., the variance diminishes. Could you please explain why such phenomenon happens?
>
> As far as we can observe, the variance does not diminish, and the method behaves in a similar way to how regular SGD does with a good choice of the learning rate, as for example 10e-2 in the case of logistic regression. We would be interested in looking into this more carefully if you could point us to an experiment/figure where this behavior with SGD happens.
>
> Thank you once more for all these constructive comments and suggested additions that allow us to improve the paper.

---

### Public Comment · ~Ricardo_Pio_Monti1 · 2017-11-07
**Nice paper!**

Dear authors,

Thank you for this paper, I really enjoyed it! :)

I have two small comments:

 - A related field which may provide additional insights in that of Adaptive filter theory [1]. A particularly relevant example would be the use of adaptive forgetting factors, where gradient information is used to tune a forgetting factor recursively.

 - A further interesting application for the proposed method could be in the context of non-stationary data. In such a setting, it may be desirable to allow the learning to rate to increase if necessary (as would be the case if, for example, the underlying data distribution changed). Potential scenarios where this could happen are streaming data applications (where model parameters are constantly updated to take into consideration new observations/drifts in the distribution) or transfer learning applications.

Best wishes and good luck!

References:
1. Adaptive Filter Theory, Simon Haykin, Prentice Hall, 2008

---

> ### Author Response · Authors · 2017-11-24
> **Thank you**
>
> Hi, both are very interesting potential applications!
>
> I think an application to non-stationary data, where the learning rate varies on the fly as new data comes in, would be very interesting indeed. We will keep this in mind.
>
> We're also looking at adaptive filter theory.
>
> Thank you very much for the pointers.

---

### Public Comment · ~Kai_Li2 · 2017-11-21
**why multiplicative adaptation is in general faster than the additive adaptation？**

 One of the practical advantages of this multiplicative
rule is that it is invariant up to rescaling and that the multiplicative adaptation is in general faster than
the additive adaptation. Why?

---

> ### Author Response · Authors · 2017-11-21
> **why multiplicative adaptation is in general faster than the additive adaptation？**
>
> You only need a logaritmic number of iterations to shift your current learning rate to another value, instead of a linear number of them. We have also seen in practice that with good hyperparameters for both implementations, the multiplicative rule adapts faster. There is also a theoretical reason that comes from the formal derivation of the rule that suggests that the multiplicative rule makes more sense than the additive one.

---

### Public Comment · ~Yi_Lian1 · 2017-12-16
**Successfully reproduced!**

This paper introduces an adaptive method to adjust the learning rate of machine learning algorithms, and aims to improve the convergence time and reduce manual tuning of learning rate. The idea is simple and straightforward: to automatically update the learning rate by performing gradient descent on the learning rate alongside the gradient descent procedure of the parameters of interest. This is achieve by introducing a new hyperparameter and specifying an initial learning rate. The idea is intuitive and the implementation is not hard.
We find that the way the experiments are set­up and described facilitates reproducibility. The data sets in the experiment are all publicly available, partitioning information of training and test data sets are clearly stated except the randomization control of training set for each experiment. Authors implemented and documented the Lua code of the proposed optimization algorithms for SGD, SGDN and Adam, and made those codes available within the torch.optim package on github. The python version of AdamHD can also be found publicly online. Since we do not have programming experience using Lua, we implemented the python version of SGDHD and SGDNHD by ourselves following the paper pseudocode, but we cannot guarantee that our implementation based on our understanding is exactly the same as the authors'. However, the code that authors used to generate the exact plots and graphs to illustrate their experiment results are not available. Thus we also implemented this part of code ourselves according to paper. Most parameters (including hyperparameters) used in experiments were given. We would suggest authors to include more hardware­specific information used to run their experiments in the paper, including time, memory, GPU and type of machine.
It is not hard to replicate the results shown in the original paper, with some effort to apply machine learning methods embedded in the Torch or PyTorch library on the given data set. Based on the results, it is great to see that most of the experiments in the study are reproducible. Specifically, the change of learning rate and training/validation loss in our replication generally follows a similar pattern to that in the paper. For example, the learning rate increases in the first few epochs in logistic regression and neural networks using SGDHD. Also, the learning rate and training/validation loss tends to oscillate starting at some point in the paper and our results shows the same pattern. However, there are also instances where the non­HD version of the optimizers perform better than the HD counterparts.
Overall, the paper is well­written, provides a promising algorithm that works at least as well as existing gradient­descent­based optimization algorithms that use a fixed global learning rate. The authors claim that an important future work is to investigate the theoretical convergence guarantees of their algorithm, which is indeed very insightful. I am hoping that the authors can also justify the theoretical support behind the adaptation of the learning rate in the sense that to what they are trying to adapt the learning rate.

---

> ### Author Response · Authors · 2017-12-29
> **Thank you**
>
> Thank you very much for your time and for reporting your results. This sort of validation is extremely valuable for us and the community.
>
> Following the decision notification, we will make a repository public with the full code in Python (including the plotting codes that we used for producing the plots in the paper). We will also add information about the hardware setup that was used for running the presented experiments.

---

### Decision · Program_Chairs · 2018-01-29
**ICLR 2018 Conference Acceptance Decision**

**Decision:**

Accept (Poster)

**Comment:**

All reviewers agreed that, despite the lack of novelty, the proposed method is sound and correctly linked to existing work. As the topic of automatically learning the stepsize is of great practical interest, I am glad to have this paper presented as a poster at ICLR.